# Trends of Phase I Clinical Trials in the Latest Ten Years across Five European Countries

**DOI:** 10.3390/ijerph192114023

**Published:** 2022-10-28

**Authors:** Davide Di Tonno, Caterina Perlin, Anna Chiara Loiacono, Luca Giordano, Laura Martena, Stefano Lagravinese, Federica Rossi, Santo Marsigliante, Michele Maffia, Andrea Falco, Prisco Piscitelli, Alessandro Miani, Susanna Esposito, Alessandro Distante, Alberto Argentiero

**Affiliations:** 1ClinOpsHub s.r.l., 72023 Mesagne, Italy; 2Department of Biological and Environmental Science and Technologies (Di.S.Te.B.A.), University of Salento, 73100 Lecce, Italy; 3Unit of Biostatistics, Epidemiology and Public Health, Department of Cardiac Thoracic Vascular Sciences and Public Health, University of Padova, Via Loredan 18, 35121 Padova, Italy; 4Department of Public Health, European University of Madrid, 28670 Madrid, Spain; 5Department of Environmental Science and Policy, University of Milan, 10123 Milan, Italy; 6Pediatric Clinic, Department of Medicine and Surgery, University of Parma, 43121 Parma, Italy; 7Euro Mediterranean Scientific Biomedical Institute (ISBEM), 72023 Mesagne, Italy; 8Centro Ricerca Clinica Salentino (CERICSAL), Veris delli Ponti Hospital, 73020 Scorrano, Italy

**Keywords:** phase 1 clinical trials, Europe, trends, ClinicalTrials.gov database

## Abstract

Background: Phase 1 clinical trials represent a critical phase of drug development because new candidate therapeutic agents are tested for the first time on humans. Therefore, international guidelines and local laws have been released to mitigate and control possible risks for human health in agreement with the declaration of Helsinki and the international Good Clinical Practice principles. Despite numerous scientific works characterizing the registered clinical trials on ClinicalTrials.gov, the main features and trends of registered phase 1 clinical trials in Europe have not been investigated. This study is aimed at assessing the features and the temporal trend of distribution of phase 1 clinical studies, carried out in the five largest European countries over a ten-year period (2012–2021), and to evaluate the impact of the Italian regulatory framework on the activation of such studies. Methods: The main data and characteristics of phase 1 clinical studies registered on the ClinicalTrials.gov database for France, Germany, Italy, Spain and the United Kingdom have been investigated and subsequently compared. The above-mentioned countries were selected based on similarities in terms of demographic and Gross Domestic Product (GDP) data available on official government websites. (3) Results: A total number of 6878 phase 1 clinical trials were registered for the five selected countries in the ClinicalTrials.gov database during the ten years analyzed; the studies were predominantly randomized (39.33%) and for-profit (76.64%). The most represented area of investigations was oncology (52.15%), followed by hematology (24.99%) and immunology (12.04%). The variability observed between the analyzed countries showed that the UK, Germany and France presented the highest reduction in the number of phase 1 clinical trials, while for Spain and Italy, a stable/increased trend was observed, although with a lower number of trials registered on the ClinicalTrials.gov database. (4) Conclusions: Italy displayed the lowest number of registered phase 1 clinical trials, even though it showed a stable trend over the years. In this regard, the Italian regulatory framework must urgently be adapted to that of other European countries (Spain has been the first country to implement the new Regulation (EU) No 536/2014) and streamline the process of clinical trial application to increase the attractiveness of the country. Moreover, nonprofit phase 1 clinical trials (which represent 19.81% of the total number of phase 1 clinical trials registered in Italy vs. 80.19% of profit phase 1 clinical studies) should be promoted and supported by the institutions, even from a financial point of view, to allow independent researchers to develop new therapeutic drugs.

## 1. Introduction

Clinical trials are essential for drug development because new drugs and/or innovative therapeutic strategies need to be tested on humans to ultimately verify their safety and efficacy in real patient settings. They typically proceed through several distinct phases (from I to IV) of variable duration and are designed to meet specific endpoints [1,2] to request therapeutics market access.

Phase 1 clinical trials represent the first step of drug development, and they are usually conducted on a small group of healthy volunteers or patients affected by a well-defined condition (e.g., infections or oncology) [3]. The primary aim of such studies is to establish the safety and tolerability of the candidate drug as well as the Maximum Tolerated Dose (MTD) of the new therapeutic agent [4,5,6,7]. Due to the intrinsic characteristics of phase 1 clinical trials, particular attention must be addressed to patients’ safety and well-being [8,9]. Indeed, a variety of guidelines and regulations have been released by different institutions to protect phase 1 clinical trial participants from possible health risks [10,11,12,13].

Moreover, particular attention is paid to preclinical data requested for phase 1 clinical trial approval due to the uncertainty of drug effects on humans [14,15], especially in the pediatric population. Indeed, such studies are strictly regulated by further guidelines that take into consideration both the scientific and ethical aspects [16,17,18] of the clinical trial protocol and the pharmaceutical characteristics of the drug.

Since new potential drugs are tested for the first time on humans through phase 1 clinical trials, and because such studies are becoming more and more complex in terms of study design [19], scientific and ethical features of phase 1 clinical trials are continuously revised to allow sponsors to be fully compliant with the current legislation [8,17,20].

To date, Directive 2001/20/EC and the new Regulation (EU) No 536/2014 represent the main challenge at the European level to create a harmonized, centralized clinical trial application process starting from a fragmented regulatory system to accelerate clinical trial approval and conduct [21,22,23]. Each EU country has approved its own laws in compliance with the above-mentioned Regulation framework set at the European level [24,25]. However, despite numerous scientific works have characterized the registered clinical trials on ClinicalTrials.gov [26,27], the main features of registered phase 1 clinical trials in Europe have not been investigated.

This study is aimed at assessing the features and the temporal trend of distribution of phase 1 clinical studies, carried out in the five largest European countries (namely France, Germany, Italy, Spain and the United Kingdom) over the 2012–2021 period and to evaluate the impact of the Italian regulatory framework on the activation of the above-mentioned studies.

## 2. Materials and Methods

### 2.1. ClinicalTrials.gov Data Set

The ClinicalTrials.gov database was reviewed. We downloaded a CSV data set (updated to 18 August 2022) comprising registered phase 1 clinical studies on ClinicalTrials.gov of five different European countries: France, Germany, Italy, Spain and the UK. The countries were selected based on similarities in terms of demographic and Gross Domestic Product (GDP) data available on official government websites.

To download data, we used the following filters: “name of the country”, “yearly phase 1”, “phase 1”, “start date of the year” and “end date of the year”.

To characterize the registered phase 1 clinical trials among the five nations and possible fluctuations during this time, a period of ten years was taken into account (from 1 January 2012 to 31 December 2021).

### 2.2. Analytical Methods

We first manually assessed the downloaded registered phase 1 clinical trials data set to obtain homogeneous data and perform the analysis. Then, we sorted out data and reported the absolute number and percentage of the temporal distribution of phase 1 clinical trials, characteristics of funding source, study phase, design and therapeutic area. The data were divided by year: from 2012 to 2021.

### 2.3. Inclusion and Exclusion Criteria for the Analysis

For the temporal distribution of phase 1 clinical trials, the absolute number of phase 1 clinical trials divided by year was reported. We represented the difference between one country and another in percentage.

As regards the funding source analysis, a different approach was used. The total number of phase 1 clinical trials funding was divided into “Industry” and “Other” per country. The latter, based on the ClinicalTrials.gov database filters, comprises individuals, universities, organizations, NIH and other U.S. federal organizations. The total number of registered phase 1 clinical trials was divided by country and the number of industry-funded studies and other-funding studies was calculated with respect to the total number of registered studies. Percentages were calculated taking into account the number of phase 1 clinical trials per country against the total number of industry or other phase 1 studies.

The percentage of early phase 1, phase 1 and phase 1/phase 2 clinical trials was calculated taking into account the number of clinical trials per phase against the total number of registered phase 1 clinical studies for each category over the ten-year period. The same approach was used for study design analysis. In this case, data were divided into “Allocation: Randomized”, “Allocation: Non-Randomized”, “Allocation: N/A” and “Other”. Data were sorted out using the above-mentioned keywords and those that did not fit in any of the categories were classified as “Other”. The percentage of “Randomized” and “Non-Randomized” phase 1 studies was calculated by taking into account the number of clinical trials per phase against the total number of studies for each category over the ten-year period. The term “Randomized” indicates a study in which the patients are divided by chance into separate groups that compare different treatments. Instead, the term “Non-Randomized” indicates a study in which the allocation is not at random.

To analyze the characteristics of the therapeutic areas, we manually searched registered data (updated to 22 August 2022) on ClinicalTrial.gov for each country.

The following filters were applied: “name of the country”, “yearly phase 1”, “phase 1”, “year start date” and “year-end date”. The “specialization name” has been written in the “other terms” field of the database.

The analysis was conducted considering 1 January 2012, as starting date, and 31 December 2021, as the ending date.

The specializations have been selected based on the 19th National Report on Clinical Trials of Medicines in Italy 2020, published by the Italian Competent Authority (Agenzia Italiana del Farmaco—AIFA) on 30 December 2020 [28]. The following areas were considered: Oncology, Hematology, Neurology, Gastroenterology, and Immunology.

Then, the percentage of each category per nation was calculated by taking into account the number of registered phase 1 studies against the total number of studies for each therapeutic area over the ten-year period.

### 2.4. Statistical Analysis

We reported absolute numbers and percentages and compared them by using the t-Student test. When it was not possible to perform the t-Student, we reported data in percentage. The difference is considered statistically significant if the *p*-value is less than 0.05.

## 3. Results

### 3.1. Temporal and Geographic Distribution of Phase 1 Clinical Trials

A total number of 6878 phase 1 clinical trials were registered from 2012 to 2021. Italy has registered 621 phase 1 clinical trials (9.03%), while the UK has registered the highest number of clinical studies among the five European countries (N = 2203; 32.03%). Germany accounted for 1588 phase 1 clinical studies (23.09%), followed by Spain (N = 1257, 18.28%) and France (N = 1209, 17.58%).

As shown in Figure 1, Italy, Spain, France and Germany have shown an increasing number of registered phase 1 clinical trials until 2016, except for the UK which showed a decreased number of registered studies from 2014 to 2016. However, the trend is followed by an increasing number of registered phase 1 studies in 2017 with a subsequent decrease until 2020. On the other hand, Italy, Spain, France and Germany have shown a fluctuation in the number of registered phase 1 clinical trials until 2020.

In addition, Italy has displayed the lowest number of phase 1 clinical trials over the years compared to the other nations, while the UK has shown the highest number of registered phase 1 clinical studies. However, it is possible to observe fluctuations within each country from 2012 to 2021 in terms of registered phase 1 clinical trials.

### 3.2. Funding Source Distribution of Phase 1 Clinical Trials

The funding source of phase 1 clinical trials across Italy, Spain, France, the UK and Germany was investigated. As it is shown in Table 1, the UK has shown the highest number (N = 1637, 31.06%) of industry-funded phase 1 clinical studies versus nonprofit phase 1 clinical trials (N = 566, 35.22%) (funded by universities, organizations, NIH or others). Germany has displayed a similar trend, with 1333 industry-funded phase 1 clinical trials (25.29%) vs. 255 nonprofit phase 1 studies (15.87%).

Italy, Spain and France have shown a higher number of phase 1 clinical trials funded by industries (N = 498, 9.45%; N = 963, 18.27%; N = 840, 15.94%, respectively) than nonprofit ones (N = 123, 7.65%; N = 294, 18.29%; N = 369, 22.96%, respectively).

Not surprisingly, the number of nonprofit phase 1 clinical studies is statistically lower than for-profit phase 1 studies (*p* = 0.008).

### 3.3. Characterization of Phase 1 Clinical Trials Based on the Study Phase

We have also taken into consideration the study phase of the registered phase 1 clinical trials across the five countries from 2012 and 2021.

Most of the studies in all countries have been pure phase 1 clinical studies, and as it is shown in Figure 2, the UK (N = 1592) and Germany (N = 1180) have displayed a higher number of pure phase 1 studies with respect to Italy (N = 289), Spain (N = 717) and France (N = 601). Indeed, in the UK the pure phase 1 clinical trials have represented 72.27% of the total registered phase 1 studies in the country, while in Germany 74.31% of the total.

Phase 1/phase 2 studies are less common than pure phase 1 clinical trials. For instance, the UK has shown fewer phase 1/phase 2 clinical studies with respect to the above-mentioned data (N = 553) representing 25.10% of the total registered phase 1 clinical trials in the nation. Germany has shown a similar trend, with 383 phase 1/phase 2 clinical trials that constitute 24.12% of the total registered phase 1 studies from 2012 to 2021 in the country. Italy has displayed similar results as pure phase 1 clinical studies (N registered phase 1/phase 2 clinical trials from 2012 to 2021 = 320, 51.53% of the total registered phase 1 clinical trials). The same trend has been observed also for Spain and France (N = 504, 40.10% and N = 461, 31.13%, respectively).

The data have also confirmed that early phase 1 clinical studies are statistically lower than pure phase 1 (*p*-value = 0.0192) and phase 1/phase 2 clinical trials in all five European countries (*p*-value = 0.0003).

### 3.4. Analysis of Phase 1 Clinical Trials by Study Design

The characteristics of phase 1 clinical trials based on study design across the five selected countries were also investigated.

As shown in Figure 3, Italy, Spain and France shared a similar trend. Indeed, the registered phase 1 clinical studies have not displayed big differences in terms of the percentage of randomized phase 1 clinical trials (Italy, 26.09%; Spain, 29.59%; France, 25.89%) and nonrandomized phase 1 clinical studies (Italy, 36.88%; Spain, 38.98%; France, 35.65%).

However, the UK and Germany showed a different trend, characterized by a higher percentage of randomized phase 1 studies (UK, 47.34%; Germany, 51.32%) rather than nonrandomized ones (UK, 25.74%; Germany, 25.50%).

### 3.5. Therapeutic Areas Distribution among Phase 1 Clinical Trials in Europe

To have a comprehensive overview of the type of phase 1 clinical trials conducted in Europe, the most frequent therapeutic areas were also investigated.

As shown in Table 2, the majority of registered phase 1 clinical trials in the EU have been conducted in the field of oncology (Italy, 54.88%; Spain, 58.36%; France, 57.70%; UK, 43.82%; Germany, 47.65%), followed by hematology (Italy, 26.56%; Spain, 22.37%; France, 22.21%; UK, 27.85%; Germany, 26.42%).

Immunological phase 1 clinical trials have been conducted with a lower frequency (Italy, 11.86%; Spain, 11.79%; France, 11.90%; UK, 11.44%; Germany, 13.57%), followed by neurological (Italy, 0.77%; Spain, 0.91%; France, 0.93%; UK, 1.40%; Germany, 2.15%) and gastroenterological phase 1 studies, which were the less frequent (Italy, 0.33%; Spain, 0.24%; France, 0.26%; UK, 0.27%; Germany, 0.72%).

## 4. Discussion

This study shows that the registered phase 1 clinical trials on ClinicalTrials.gov during the last ten years are predominantly randomized (39.33%), for profit (76.64%), oncological (52.15%) and hematological (24.99%) clinical trials, even though methodological aspects greatly vary among them. Indeed, France, Germany, Italy, Spain and the UK shared a similar distribution of phase 1 clinical trials in terms of study design, study phase and funding source.

As it is possible to understand from the collected data, the UK has displayed the highest number of registered phase 1 clinical trials over the ten-year period. Data suggest the fact that British professionals working in the clinical research field displayed the ability to successfully activate a considerable number of phase 1 clinical trials.

However, it is necessary to carry out an in depth analysis of the impact that Italian legislation has had on the activation of such studies.

Indeed, the Italian Competent Authority (Agenzia Italiana del Farmaco—AIFA) published the AIFA Determination No 809 in 2015 that established the minimum requirements that a phase 1 clinical unit or laboratory must have to conduct phase 1 clinical trials. The Determination has become effective in 2016.

The document established high-quality standards and clinical units have had to adapt their context in order to improve the overall quality of phase 1 clinical trials conducted in Italy. As it is possible to understand from the analysis of the collected data, it is reasonable to conclude that the determination did not have a negative impact on the number of registered phase 1 clinical studies after its implementation, as the number of registered studies remained constant during the years with minimal fluctuations.

Hence, the AIFA Determination ensures that phase 1 clinical trials are managed following high quality standards to protect the safety and well-being of enrolled patients and data credibility, as per Good Clinical Practice and the Declaration of Helsinki.

However, the total number of registered phase 1 clinical trials in Italy is lower than in the other selected European countries. This aspect can be attributed to several factors, among them the failure to implement the new Regulation (EU) No 536/2014 might be the most relevant. Indeed, few implementing decrees have been released by the Italian Government. Thus, it becomes urgent to adapt the Italian regulatory framework to those of the other European countries in order to increase the attractiveness and number of approved phase 1 clinical trials. Indeed, Spain has become one of the first nations to have implemented the new Regulation (EU) No 536/2014, showing a higher number of registered phase 1 clinical trials. Thus, it is reasonable that a unique clinical trial application may accelerate the activation of phase 1 clinical studies.

In contrast, the IEC’s effectiveness in clinical trials approval should be considered, given the large number of IECs located in Italy and the heterogeneity of the submission process of clinical trials-related documents. De facto, the new Regulation (EU) No 536/2014 aims to optimize the clinical trial application process by increasing the latter in terms of efficiency and rapidity. In line with that purpose, the new Regulation (EU) reduces the amount of IECs, even though they are allowed to organize their activities based on local legislation. Despite the freedom given by the new Regulation (EU) No 536/2014, no implementing decrees have been released by the Italian authorities, thus delaying the IECs reorganization in view of the effective entry into force of the new Regulation (EU) No 536/2014 in 2023.

Based on the current situation and the Spanish model, the new Regulation (EU) No 536/2014 implementation should be considered an advantage in terms of competitiveness by the Italian government. Indeed, after the implementation of the new Regulation (EU) No 536/2014, Spain registered 846 phase 1 clinical trials from 2016 to 2021 versus 383 registered phase 1 studies in Italy. This opportunity may lead to the reduction in bureaucracy and an acceleration of the clinical trial approval period, resulting in an increased attractiveness of Italy at a European level, even from a financial point of view [29,30,31,32].

Furthermore, more attention should be given to nonprofit phase 1 clinical studies in order to give scientists the possibility to develop new and alternative therapeutics. Independent phase 1 clinical studies should be appropriately promoted and supported by the institutions [23]. In fact, in Italy, they represent only 19.81% of the total number of phase 1 clinical trials registered versus 80.19% of profit phase 1 clinical studies.

This aspect should be carefully taken into consideration by the regulatory authorities and governments in order to promote academic-based phase 1 clinical studies as much as possible. For instance, difficulties may derive from a heterogenous regulatory framework that does not fully consider the issues related to the clinical trial application among the countries and health insurance costs for academic researchers.

The impact of a harmonized regulatory system on the number of registered phase 1 clinical trials should be investigated in the future, after the entry into force of the new Regulation (EU) No 536/2014.

Despite the innovative research conducted, further investigations are needed to better understand the overall trend of registered phase 1 clinical trials in the EU.

Gender and phase 1 clinical study enrollment trends should be taken into consideration to provide a better insight into gender equity and accessibility to innovative therapeutic agents for the general population, as well as for children and the elderly, which will surely benefit from new commercialized drugs [33].

The results of this study represent part of the European Union, as it has considered only five nations that displayed similarities in terms of demographic and Gross Domestic Product (GDP) data available on official government websites. Thus, this pilot study should encourage the accurate analyzing of the general trend of registered phase 1 clinical trials in other European Countries in order to gain a more complete picture of the registered phase 1 clinical trials’ characteristics and distribution in the EU.

Moreover, further studies must be conducted to determine the most predominant therapeutic areas in Europe. This study only considered five specializations based on the 19th National Report on Clinical Trials of Medicines in Italy 2020 published by AIFA on 30 December 2020. Even though they provided a general overview of phase 1 clinical study status, some difficulties have been encountered during data analysis in relation to clinical trial classification. This is due to the fact that more and more clinical studies show an overlap of different therapeutic areas (e.g., oncology and hematology areas), and it is reasonable to think that this trend may increase in the future. For example, hematological clinical studies may also include liquid cancer such as leukemia. In this regard, the above-mentioned consideration is in line with the study phase analysis, which indicates an increasing number of phase 1/phase 2 clinical trials.

For this reason, a more precise classification of registered clinical trials on ClinicalTrials.gov might be essential for correct data analysis.

Moreover, this study does not consider phase 1 clinical trials of Advanced Therapy Medicinal Products (ATMPs) [34]. The latter is growing fast, and in depth studies are essential to better understand the current and future trends in order to facilitate their commercialization and availability, including their reimbursement by the national Competent Authorities [35].

## 5. Conclusions

Our research demonstrates that Italy shows the lowest number of registered phase 1 clinical trials compared to Spain, France, Germany and the UK, even though it displayed a stable trend over the ten years.

Data suggest that Italy has the potential to conduct numerous phase 1 clinical trials like other European countries. For this reason, some limitations must be overcome. It becomes urgent to adapt the national regulatory framework to that of other European countries and try to streamline the process of clinical trial application to increase the attractiveness of the country. Furthermore, nonprofit phase 1 clinical trials should be promoted and supported by the institutions, even from a financial point of view, to allow independent researchers to develop new therapeutic drugs.

## Figures and Tables

**Figure 1 ijerph-19-14023-f001:**
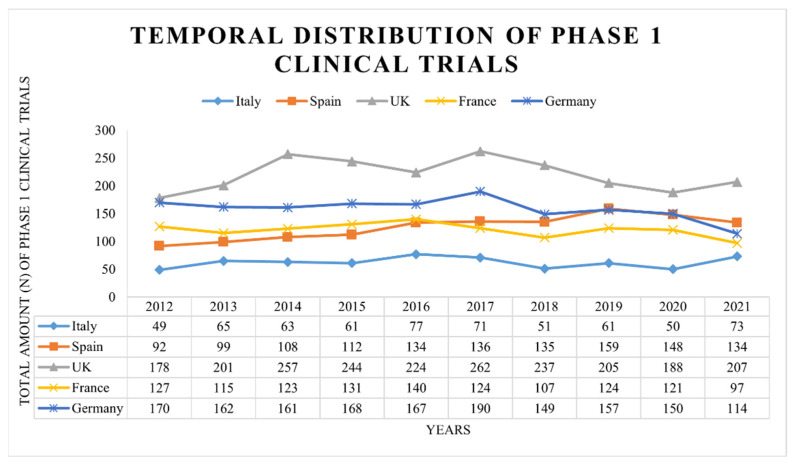
Temporal distribution of phase 1 clinical trials. The total number of registered phase 1 clinical trials on ClinicalTrials.gov was divided by year, from 2012 to 2021. Five European countries were selected: France, Germany, Italy, Spain and the UK.

**Figure 2 ijerph-19-14023-f002:**
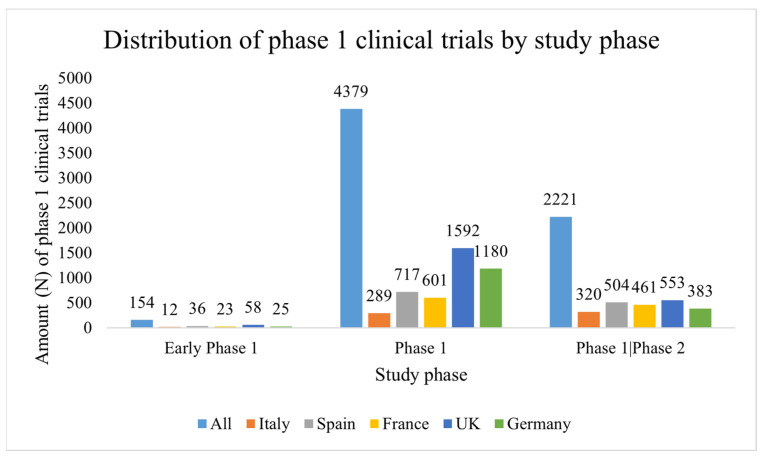
Distribution of phase 1 clinical trials by study phase. The total number of registered phase 1 clinical trials on ClinicalTrials.gov was divided by country and study phase. Five European countries were selected: France, Germany, Italy, Spain and the UK. The period from 1 January 2012 to 31 December 2021 was taken into consideration. Early phase 1 vs. phase 1: *p*-value = 0.0192; Early phase 1 vs. phase 1/phase 2: *p*-value = 0.0003.

**Figure 3 ijerph-19-14023-f003:**
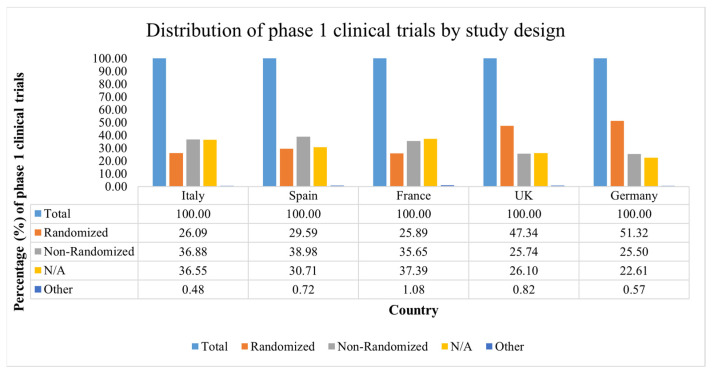
Distribution of phase 1 clinical trials by study design. The total number of registered phase 1 clinical trials on ClinicalTrials.gov was divided by country and study design. Five European countries were selected: France, Germany, Italy, Spain and the UK. The period from 1 January 2012 to 31 December 2021 was taken into consideration.

**Table 1 ijerph-19-14023-t001:** Funding source of phase 1 clinical trials in EU. Five European countries were selected: France, Germany, Italy, Spain and the UK. The period from 1 January 2012 to 31 December 2021 was taken into consideration.

Funding Source	Industry		Other	
	(N)	(%)	(N)	(%)
Italy	498	9.45	123	7.65
Spain	963	18.27	294	18.29
France	840	15.94	369	22.96
Germany	1333	25.29	255	15.87
UK	1637	31.06	566	35.22
All	5271	100	1607	100

**Table 2 ijerph-19-14023-t002:** Therapeutic areas distribution among phase 1 clinical trials. The total number of registered phase 1 clinical trials on ClinicalTrials.gov was divided by country and therapeutic area. Five European countries were selected: France, Germany, Italy, Spain and the UK. The period from 1 January 2012 to 31 December 2021 was taken into consideration.

Country	All	Oncology	Hematology	Neurology	Gastroenterology	Immunology
Italy	911 (100%)	500 (54.88%)	242 (26.56%)	7 (0.77%)	3 (0.33%)	108 (11.86%)
Spain	1645 (100%)	960 (58.36%)	368 (22.37%)	15 (0.91%)	4 (0.24%)	194 (11.79%)
France	1513 (100%)	873 (57.70%)	336 (22.21%)	14 (0.93%)	4 (0.26%)	180 (11.90%)
UK	1853 (100%)	812 (43.82%)	516 (27.85%)	26 (1.40%)	5 (0.27%)	212 (11.44%)
Germany	1253 (100%)	597 (47.65%)	331 (26.42%)	27 (2.15%)	9 (0.72%)	170 (13.57%)

## Data Availability

This study is based on a public database. The raw data supporting the findings of this study will be provided upon request to the corresponding author.

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
