# Peer review of "Trends of Phase I Clinical Trials in the Latest Ten Years across Five European Countries"

_ijerph, 2022, doi:10.3390/ijerph192114023_

Round 1

Reviewer 1 Report

The manuscript titled: “Trends of Phase I Clinical Trials in the Latest Ten Years across Five European Countries” is very interesting for readers and well written. Despite of that, the presented results was discussed not objectively. The Authors have written: As shown in figure 1, Italy, Spain, France, and Germany have shown an increasing number of registered phase 1 clinical trials until 2016, except for the UK. The fluctuations in the amount of phase I clinical trial are characterised in each above mentioned countries. The trend is followed by a slight increase until 2020, what was observed in both, UK and Italy. The next figures shows a similar trends in all countries, so the comparison of Italy to other countries in conclusions is proper. The conclusions are adequate to presented results.

Major points:

- in discussion section,  please underline the importance of UK in clinical trials.

Author Response

Respected Reviewer 1, I thank you for your comments and for encouraging us to improve the quality of the submitted paper. Please find below the answers.

Point 1: The manuscript titled: “Trends of Phase I Clinical Trials in the Latest Ten Years across Five European Countries” is very interesting for readers and well written. Despite of that, the presented results was discussed not objectively. The Authors have written: As shown in figure 1, Italy, Spain, France, and Germany have shown an increasing number of registered phase 1 clinical trials until 2016, except for the UK. The fluctuations in the amount of phase I clinical trial are characterised in each above mentioned countries. The trend is followed by a slight increase until 2020, what was observed in both, UK and Italy. The next figures shows a similar trends in all countries, so the comparison of Italy to other countries in conclusions is proper. The conclusions are adequate to presented results.

Major points:

- in discussion section, please underline the importance of UK in clinical trials.

Response 1: the importance of UK in clinical trials conduct has been added in the discussion section. In particular, the ability of successfully manage clinical trials in the british country was emphasized (lines 425-428).

Reviewer 2 Report

This is an interesting retrospective and analysis-based clinical phase 1 trial-relevant article for the clinical trial phase 1 trends in five major European countries (Italy, Spain, France, Germany and UK) in the past 10 years (2012 to 2021). The methodology part including phase 1 clinical trial data retrieving, analytical parameters, inclusion/exclusion criteria for the analysis were succinctly described. English writing is basically good enough for publication.

Questions

1. Based on this reviewer’s understanding, phase 1 clinical trials in oncology area were single arm with open label. So, the question is what is the definition of “randomized” and “non-randomized”? It is not clear based on the current methods provided. Suggestions: The definition of these and other terms may be defined in the methodology part.

2. Whether the hematology clinical trials include or exclude the liquid cancer like leukemia? Suggestions: This may need a clarification when define these terms in methodology part or elsewhere.

Additional suggestions:

3. All figures may consider using color to be clearer and straightforward, if only electronic version of this journal exists (in this case, color figures will not have more expenses) or use pattern if hard copy still is this journal format.

4. For Figure 1, the authors may consider making a population size of each country-normalized to the numbers of clinical trials in each of the five countries to see if the current big differences of the trial numbers in different countries could be decreased.

5. The description of “As it is shown in figure 2, the UK has shown the highest num-148 ber (N= 1637, 31.06%) of industry-funded phase 1 clinical studies versus 566 non-profit 149 phase 1 clinical trials (35.22%) (funded by universities, organizations, NIH or others).” may need some clarity revision, in order to let the audiences/readers easy to understand the second percentage (35.22%) calculated (based on which number). In other words, why (N= 1637, 31.06%) versus N= 149 could be 35.22%?  Similar situation/issue existed for Germany case “1333 industry funded-phase 1 clinical trials 151 (25.29%) vs. 255 non-profit phase 1 studies (15.87%)” as well as the description of “Italy, Spain, and France have shown a higher number of phase 1 clinical trials funded 153 by industries (N= 498, 9.45%; N=963, 18.27%; N=840, 15.94% respectively) than non-profit 154 ones (N=123, 7.65%; N=294, 18.29%; N=369, 22.96% respectively).”. Basically, it is not easy to follow the percentage logic.

6. One way to resolve the above clarification, whether it is possible for these authors to consider adding the methodological description with the data calculation for clarification of the percentage calculation.

7. For the bar figures, these authors may consider putting the number on the top of each bar.

8. Finally, these authors may also consider using Tables for the data presentation, which may have advantages over the text description for easy understanding.

Author Response

Estimated Reviewer 2, thank you for your comments and suggestions. Please find below the answers.

Point 1: Based on this reviewer’s understanding, phase 1 clinical trials in oncology area were single arm with open label. So, the question is what is the definition of “randomized” and “non-randomized”? It is not clear based on the current methods provided. Suggestions: The definition of these and other terms may be defined in the methodology part.

Response 1: in order to better clarify the differences between the terms “Randomized” and “Non-Randomized”, the definitions have been added in the material and methods section (lines 137-140).

Point 2: Whether the hematology clinical trials include or exclude the liquid cancer like leukemia? Suggestions: This may need a clarification when define these terms in methodology part or elsewhere.

Response 2: the hematology clinical trials may include liquid cancers as leukemia. Indeed, based on the ClinicalTrials.gov desing, no filters to distinguish such diseases are now available. For this reason, in order to avoid bias during data analysis, the manual clinical trial classification among the selected therapeutic areas was not considered. In this regard, we looked at the registered phase 1 clinical trials by adding the “therapeutic name” in the “Other terms” field. Therefore, we highlighted in the discussion section the possibility that liquid cancers may be included in the hematology clinical trials classification (lines 517-519).

Additional suggestions:

Point 3: All figures may consider using color to be clearer and straightforward, if only electronic version of this journal exists (in this case, color figures will not have more expenses) or use pattern if hard copy still is this journal format.

Response 3: We thank you for this consideration. All figures of the article have been changed using color to be clearer and straightforward.

Point 4: For Figure 1, the authors may consider making a population size of each country-normalized to the numbers of clinical trials in each of the five countries to see if the current big differences of the trial numbers in different countries could be decreased.

Response 4: we thank you the reviewer for this additional suggestion. We are aware that more analysis need to be carried out. However, this is considered a pilot study that should encourage to perfom in-depth studies and provide a clearer pictures of the current registered phase 1 clinical trials status in EU (lines 506-509).

Point 5: The description of “As it is shown in figure 2, the UK has shown the highest num-148 ber (N= 1637, 31.06%) of industry-funded phase 1 clinical studies versus 566 non-profit 149 phase 1 clinical trials (35.22%) (funded by universities, organizations, NIH or others).” may need some clarity revision, in order to let the audiences/readers easy to understand the second percentage (35.22%) calculated (based on which number). In other words, why (N= 1637, 31.06%) versus N= 149 could be 35.22%?  Similar situation/issue existed for Germany case “1333 industry funded-phase 1 clinical trials 151 (25.29%) vs. 255 non-profit phase 1 studies (15.87%)” as well as the description of “Italy, Spain, and France have shown a higher number of phase 1 clinical trials funded 153 by industries (N= 498, 9.45%; N=963, 18.27%; N=840, 15.94% respectively) than non-profit 154 ones (N=123, 7.65%; N=294, 18.29%; N=369, 22.96% respectively).”. Basically, it is not easy to follow the percentage logic.

Response 5: We thank you for this consideration. To make the paragraph easier to understand, the latter has been modified to allow the reader following the logic of the data analysis.

Point 6: One way to resolve the above clarification, whether it is possible for these authors to consider adding the methodological description with the data calculation for clarification of the percentage calculation.

Response 6: please consider the answer of point 5 (lines 124-128).

Point 7: For the bar figures, these authors may consider putting the number on the top of each bar.

Response 7: To improve the overall quality of reported figures, the number on the top of each bar was added. In figures where it was not possible to implement such modification due to graphical representation, a table below the bar plot was included.

Point 8: Finally, these authors may also consider using Tables for the data presentation, which may have advantages over the text description for easy understanding.

Response 8: Tables were added in figures 1 and 3. Moreover, the figures 2 and 5 have been substituted with tables 1 and 2.

Reviewer 3 Report

This is a valuable paper that reviews the state of Phase I trials in Europe and discusses issues related to the expansion of Phase I trials. However, the following concerns should be addressed in this original paper with additional data.

Major comments:
Comment 1:
The authors state that ”However, despite numerous scientific works have characterized the registered clinical trials on ClinicalTrials.gov [26,27], the main features of registered phase 1 clinical trials in Europe are still unclear” (page 2, line 71-73), this study analyzed European data registered on ClinicalTrials.gov. Please specify in the Introduction why the European data could not be analyzed in the previous study. Also, this study focused on 5 European countries, but there are 27 countries belonging to the EU. Please clearly state in the Introduction why you selected these 5 countries for this study. 

Comment 2:
The paper has achieved its stated purpose “This study is aimed at assessing the number and temporal trend of distribution of phase 1 clinical studies carried out over a ten-year period (2012-2021) in the five biggest European countries, namely France, Germany, Italy, Spain and the United Kingdom” and has achieved its purpose. However, there is a lack of explanation as to what the purpose of the study was. Since the paper is not only descriptive statistical data but also a comparative study, there must be a reason for conducting this study, so please add the purpose. The authors must have some hypothesis or purpose, especially since they selected five countries for this comparative study.

Comment 3:
In the conclusion section, the authors cite reasons for Italy's low enrollment in the Phase I study compared to the other four countries compared, including the country's regulatory framework not matching that of other European countries, the approval process, and financial issues. However, since the only data supporting this is the financial data found in the said paper, other data should be added to the discussion. Similarly, there is no data on the regulatory framework or clinical trial application process in the abstract, so this data needs to be added to the Results section of the abstract.

Author Response

Respected Reviewer 3, thank you for your comments and suggestions that have helped us to deepen the analysis. Please find below the answers.

This is a valuable paper that reviews the state of Phase I trials in Europe and discusses issues related to the expansion of Phase I trials. However, the following concerns should be addressed in this original paper with additional data.

Point 1: The authors state that ”However, despite numerous scientific works have characterized the registered clinical trials on ClinicalTrials.gov [26,27], the main features of registered phase 1 clinical trials in Europe are still unclear” (page 2, line 71-73), this study analyzed European data registered on ClinicalTrials.gov. Please specify in the Introduction why the European data could not be analyzed in the previous study. Also, this study focused on 5 European countries, but there are 27 countries belonging to the EU. Please clearly state in the Introduction why you selected these 5 countries for this study.

Response 1: We thank you the reviewer for the comment. The abstract section has been modified to clearly state that the main features of phase 1 clinical trials in Europe have not been investigated (lines 24-25). Moreover, we have specified that the five European countries were selected based on similarities on demographic and Gross Domestic Product (GDP) data available on official government websites (lines 30-32). The specification has been also highlighted in the materials and methods section (lines 104-105).

Point 2: The paper has achieved its stated purpose “This study is aimed at assessing the number and temporal trend of distribution of phase 1 clinical studies carried out over a ten-year period (2012-2021) in the five biggest European countries, namely France, Germany, Italy, Spain and the United Kingdom” and has achieved its purpose. However, there is a lack of explanation as to what the purpose of the study was. Since the paper is not only descriptive statistical data but also a comparative study, there must be a reason for conducting this study, so please add the purpose. The authors must have some hypothesis or purpose, especially since they selected five countries for this comparative study.

Response 2: We thank you for the observation. The research has been conducted to assess the trend of phase 1 clinical trials over a ten-year period in the five mentioned European countries and to evaluate the impact of the Italian regulatory framework on the registration (and activation) of phase 1 clinical studies. Therefore, the scope of the article has been highlighted in the abstract section (lines 26-27), in the introduction (lines 84-88) and in the discussion (lines 367-368 and lines 436-439).

Point 3: In the conclusion section, the authors cite reasons for Italy's low enrollment in the Phase I study compared to the other four countries compared, including the country's regulatory framework not matching that of other European countries, the approval process, and financial issues. However, since the only data supporting this is the financial data found in the said paper, other data should be added to the discussion. Similarly, there is no data on the regulatory framework or clinical trial application process in the abstract, so this data needs to be added to the Results section of the abstract.

Response 3: We thank you the reviewer for the comment that helps improving the overall quality of the article. In the discussion section, we have concluded that the Italian regulatory framework must match that of the other European countries because of the imminent entry in force of the new Regulation (EU) No 536/2014. The existence of a harmonized application system should accelerate phase 1 clinical trials activation and therefore increase the actractiveness of the country. Indeed, Spain can be considered as a model case-study because it has been one of the first countries to have immediately implemented the new Regulation and has shown higher number of registered phase 1 clinical trials (lines 466-468) compared to Italy. Hypothesis reported in the scientific article are also in line with data found in the literature and cited in the article itself (ref 21: Giannuzzi V, Altavilla A, Ruggieri L, Ceci A. Clinical Trial Application in Europe: What Will Change with the New Regulation? Sci Eng Ethics. 2016 Apr;22(2):451-66. doi: 10.1007/s11948-015-9662-0. Epub 2015 Jun 3. PMID: 26037896.; ref 23: De Feo G, Frontini L, Rota S, Pepe A, Signoriello S, Labianca R, Sobrero A, De Placido S, Perrone F. Time required to start multicentre clinical trials within the Italian Medicine Agency programme of support for independent research. J Med Ethics. 2015 Oct;41(10):799-803. doi: 10.1136/medethics-2012-100803. Epub 2015 Jun 11. PMID: 26066362.; ref 29: Petrini C, Brusaferro S. Ethics committees and research in Italy: seeking new regulatory frameworks (with a look at the past). Commentary. Ann Ist Super Sanita. 2019 Oct-Dec;55(4):314-318. doi: 10.4415/ANN_19_04_02. PMID: 31850856.). However, we are aware that one limitation of the article is the ability to evaluate the efficacy of the new system because it has not been implemented yet. Indeed, we reported in the discussion that further studies are necessary to better understand the advantages of the new Regulation (lines 495-497).

As regards the data supporting the financial issues, authors have concluded that non-profit phase 1 clinical trials should be supported by the Italian institutions because they represent only 19.81% of the total registered studies in Italy over the ten-years period. To support such statement, data were added in the bastract section (lines 44-45) and in the discussion (lines 488-489).

Reviewer 4 Report

The abstract is a little hard to follow related to line 32-35 relating to Italy--the text helps in understanding and the conclusion is fine.  The reader does not have enough context to understand the recommendation for Italy in the abstract.

Avoid using lastly in all of the text.  page 3 line 99-100 missing a "to"

Line 100 remove lastly

2.4 stats, why t-test  (student t-test) --should only be used for comparison of percentages not absolute numbers (categorical, ordinal in nature), please make clear.  

page 6 lines 219-223  Are the % among the countries that significantly different  81% to 71%, how many trials would that number entail?  I look at that and think it would be okay.  

Page 7 line 259 eliminate as a matter of fact

Page 7 line 261 remove in line this

page 8 line 294 remove First

page 8 line 298 remove secondly

page 8 line 312 remove lastly

Interesting study

Author Response

Estimated Reviewer 4, thank you for your comments and suggestions that have helped to improve the quality of data presented in the article. Please find below the answers.

Point 1: Avoid using lastly in all of the text. Page 3 line 99-100 missing a "to". Page 7 line 259 eliminate as a matter of fact. Page 7 line 261 remove in line this. Page 8 line 294 remove First. Page 8 line 298 remove secondly. Page 8 line 312 remove lastly. Line 100 remove lastly

Response 1: The text has been modified to avoid “lastly” in the text and add “to” (115-116). Moreover, the following words have been eliminated: “as a matter of fact” (line 450), “in line this” (line 452), “First” (line 502), “secondly” (line 506), “lastly” (lines 100 and 525). Moreover, a further English check has been performed.

Point 2: 2.4 stats, why t-test  (student t-test) --should only be used for comparison of percentages not absolute numbers (categorical, ordinal in nature), please make clear. 

Response 2: The 2.4 paragraph has been modified to make clear the use of t-test.

Point 3: page 6 lines 219-223  Are the % among the countries that significantly different  81% to 71%, how many trials would that number entail?  I look at that and think it would be okay. 

Response 3: To clarify the features of registered phase 1 clinical studies therapeutic areas, the text of the paragraph 3.5 has been modified by adding the table 1, containing both amount of phase 1 clinical studies and related percetages.

Round 2

Reviewer 1 Report

I recommend this manuscript for publication.

Reviewer 3 Report

Authors provide appropriate revisions in response to peer review comments. I believe that the paper is worthy of publication in the journal.